# Force Prediction and Material Removal Mechanism Analysis of Milling SiCp/2009Al

**DOI:** 10.3390/mi13101687

**Published:** 2022-10-07

**Authors:** Rong Wang, Man Zhao, Jian Mao, Steven Y. Liang

**Affiliations:** 1School of Mechanical and Automotive Engineering, Shanghai University of Engineering Science, Shanghai 201620, China; 2School of Mechanical Engineering, Georgia Institute of Technology, North Ave NW, Atlanta, GA 30332, USA

**Keywords:** SiCp/Al composites, milling parameters, milling force, removal mechanism

## Abstract

In recent years, medium- and low-volume fraction silicon carbide particle-reinforced aluminum matrix composites (SiCp/Al) have increasingly become a key material in the aerospace industry. Force prediction and material removal mechanism analysis of milling SiCp/Al are necessary to improve the surface integrity of products. An orthogonal experiment of SiCp/2009Al with a volume fraction of 20% was carried out, and the effect of the milling parameters on milling force was studied with the input parameters of milling speed, feed rate, and milling depth. Thereby, the empirical force model of milling SiCp/2009Al is established by fitting the experiential data based on the multiple linear regression analysis methods. Moreover, the effects of the milling parameters on the force were analyzed. Finally, the material removal mechanism of milling SiCp/Al is analyzed based on dislocation theory. The analyzed results reveal that the removal mechanism of the SiCp/Al composites includes plastic deformation of the aluminum matrix, cutting of particles, fragmentation, and deboning. Based on dislocation theory and maximum undeformed thickness theory, the effect of cutting parameters on the form of material removal was analyzed, which serves as a guide for selecting appropriate machining parameters to obtain improved machining quality of SiCp/Al composites.

## 1. Introduction

Aluminum matrix composites with silicon carbide reinforcement (SiCp/Al) have characteristics of both metals and nonmetals [1]. They have characteristics that include high specific strength, high specific modulus, wear resistance, strong electrical and thermal conductivity, radiation resistance, and a low coefficient of thermal expansion when compared to common metals. In wealthy nations that are vying for technological advantages, this material has emerged as one of the hottest topics [2]. The domains of aerospace, automotive, electronics and electrical appliances, optical instruments, and other civic industries will all gradually transition away from some traditional metallic materials [3,4]. Medium- and low-volume fraction SiCp/Al composites with both metallic plasticity and non-metallic high-strength qualities are employed in industry more frequently than high volume fraction, aluminum-based SiC composites [5,6].

Even though SiCp/Al composites are becoming more and more popular and look promising, they are hard to work with because they have SiC particles in them. Cutting force is a key part of the machining process. It can directly affect how well the material is machined and how well it is processed [7]. So, looking at the milling force of SiCp/Al composites can help us figure out how milling works and improve the accuracy of machining.

Cutting force has been widely discussed and explored by scholars. Pramanik et al. [8] proposed a cutting force model for SiCp/Al composites and explained that the mechanism of force generation is due to three factors: chip formation force, plowing force, and particle fracture force. The chip-forming force was obtained by Merchant, the plow-shaped deformation of the substrate was calculated using the slip-line plastic field theory, and the chip-forming force caused by particle fracture was finally calculated using the Griffith fracture theory. Kishawy et al. [9] conducted orthogonal cutting experiments on aluminum matrix composites with different volume fractions under different cutting parameters and established a cutting force prediction model based on the energy method. Duan et al. [10] conducted cutting experiments on SiCp/Al composites with a volume fraction of 50% to explore the influence of the cutting parameters on cutting force and studied the relationship between chip formation and cutting force. Jeyakumar et al. [11] established a prediction model of cutting force, machined surface quality, and tool wear amount when machining Al6061/SiC composites with carbide endmills based on the surface response method, and they were in good agreement with the experimental values. Dabade et al. [12] developed a model for predicting the turning force of SiCp/Al composites and considered the effect of volume fraction, SiC particle size, cutting parameters, and tool parameters on cutting force and machined surface roughness. Han et al. [13] explained the effects of cutting force, cutting amount, content, and SiC particle size on the reinforced matrix through cutting experiments of SiCp/Al2024 composites and found that the depth of the cut and feed components were greater than the main cutting force. The phenomenon will appear when cutting with K-type carbide tools when the volume fraction of the strengthening matrix is above 18%.

Since the SiCp/Al composite material is made up of two phases, researchers are also trying to figure out how to remove it. There are two main ways to remove material: brittle fracture and plastic forming. [14]. Brittle materials are removed by voids and cracks or, by extension, spalling and breaking [15], while plasticity is nowhere similar to the chip formation process in grinding, which involves sliding, plowing, and chip forming. So there are three different forms of removal for this material: brittle removal, plastic removal, and brittle–plastic transition [16]. Du et al. [17] proposed the removal mechanism of particles when machining SiCp/Al composites with diamond abrasive tools. The study found that the body has a large plastic deformation, so the aluminum mixed with excess SiC particles is cut off from the surface. SiC particles can be removed in a variety of ways, such as crushing, fractures, microcracks, shearing, and pulling. Zha et al. [18] studied the removal mechanism of SiCp/Al composites under rotating ultrasonic vibration through composite ultrasonic vibration-assisted scratching experiments. The study showed that the particle removal method plays a decisive role in the formation of the machined surface.

As seen from the above studies, some empirical models of the cutting force of SiCp/Al composites have been studied based on experiments to observe the effect of the experimental parameters on the cutting force. Few scholars have investigated the material removal mechanism by analyzing the milling forces. In this paper, milling experiments of SiCp/Al composites were conducted with a volume fraction of 20%, and the effects of the milling parameters (milling speed, feed rate, and milling depth) on the milling force were studied. To analyze the impact of the milling force on the material removal mechanism, a SiCp/Al composite milling force prediction model was established by multiple regression analysis and validated by data. Then, the removal mechanism of matrix and silicon carbide particles during the milling of silicon carbide composites is analyzed according to the theory of dislocation and the theory of maximum unformed thickness. The maximum thickness of the undeformed chip and the critical cutting thickness of the material is analyzed theoretically. The brittle-plastic removal of particles can be controlled by controlling milling parameters to achieve good surface quality.

## 2. Experimental Conditions

### 2.1. Milling Experiments 

The workpiece material of the milling experiment was SiCp/2009Al with a volume fraction of 20%, the size of the reinforcing particles was 5–7 μm, the raw material was a plate, the size was 180 × 45 × 15 mm, the heat treatment state was T4, the raw material was processed by wire cutting, and processed into a milling specimen with a size of 30 × 30 × 19 mm. The basic performance parameters of the material are shown in Table 1. The experiment was carried out on a vertical milling machine tool, as shown in Figure 1a,b, using a solid carbide steel milling cutter, with a diameter of *D* = 10 mm, and the number of teeth being *Z* = 4. The mechanical performance parameters of the tool are shown in Table 2. Due to the high hardness of SiC particles, it is easy to cause tool wear during processing. To reduce the tool wear rate [16], this experiment was processed in a dry condition with no coolant. The milling force signal acquisition was carried out using the milling force measurement system of the Kistler Company in Switzerland, which includes a dynamometer, 9272 four-component piezoelectric cutting dynamometer, HR-CA type charge amplifier, HR-DAQ-1212M type data acquisition device, and a computer.

### 2.2. Experimental Design

In the process of machining, the milling force is affected by many factors. In this paper, an orthogonal experimental analysis of three factors and four levels is designed from the influence of milling speed, feed rate, and milling depth on the three-way milling force. The orthogonal design can greatly reduce the number of experiments. The number of times does not reduce the reliability of the experiment. The detailed parameters of the experiment are shown in Table 3, which are selected based on the milling parameters for machining load-bearing components in aerospace (e.g., Mars wheels, drill rods, etc.).

In order to expand the number of orthogonal experimental groups, blank columns were set for the orthogonal experiments. The traditional orthogonal experiment was expanded to 32 groups using SPSS software, and the experimental protocol shown in Table 4 was obtained. Table 4 contains 32 sets of data to perform the milling force analysis. Nine sets of data, namely, 2nd, 7th, 14th, 15th, 16th, 22nd, 25th, 28th, and 30th, were selected for model validation, and the remaining data were used for model fitting.

## 3. SiCp/Al Milling Force Analysis

### 3.1. Experiment Results

As shown in Figure 2, in the milling force experiment, the measured milling force has three components: the milling force $F_{x}$ in the *x*-axis direction, the milling force $F_{y}$ in the *y*-axis direction, and the milling force $F_{z}$ in the *z*-axis direction [19].

Due to the large fluctuation of the milling force in and out of the tool, the stable stage is selected for analysis. To eliminate the influence of spindle vibration on the milling force, low-frequency filtering is performed on the milling force signal, as shown in Figure 2. The magnitude of the milling force is obtained by averaging the absolute values of 50 peak points. To ensure the accuracy of the experimental data, each group of experiments is repeated twice to obtain the average value.

### 3.2. Force Prediction Model Establishment and Verification 

Based on the multiple linear regression analysis methods, the empirical force model of milling SiCp/2009Al milling force is established by fitting the experiential data, which characterizes the relationship between the influencing factors and the milling force in the milling process. The milling force formula in this paper is assumed to be:(1)$$F=C_{F}\times {v_{c}}^{a}\times f^{b}\times {a_{p}}^{c}$$
where, $C_{F}$ represents the correction coefficient of the model, which is related to the geometric parameters of the tool and the assembly angle, $v_{c}$ represents the milling speed, *f* represents the feed rate, and $a_{p}$ represents the milling depth. The unknown parameters of a,b, and c represent the indices related to the milling speed, feed rate, and milling depth, which are the relevant quantities required for fitting.

Equation (1) is in the form of an exponential equation, and the coefficients of the fitting equation are usually fitted to a linear equation, so take the logarithm on both sides of the equal sign of Equation (1) to change it into a linear function, namely:(2)$$\mathrm{lg}\left(F\right)=\mathrm{lg}(C_{F})+a\cdot \mathrm{lg}\left(v_{c}\right)+b\cdot \mathrm{lg}\left(f\right)+c\cdot \mathrm{lg}\left(a_{p}\right)\;$$
let P=lg(F), $\alpha_{0}=\mathrm{lg}\left(C_{F}\right)$, $\alpha_{1}=\mathrm{lg}\left(v_{c}\right)$, $\alpha_{2}=\mathrm{lg}\left(f\right)$, $\alpha_{3}=\mathrm{lg}\left(a_{p}\right)$, then the above equation becomes the general expression of standard linear multiple regression:(3)$$P=\alpha_{0}+a\cdot \alpha_{1}+b\cdot \alpha_{2}+c\cdot \alpha_{3}\;$$

Based on the multiple linear regression analysis methods, the regression parameters a, b, and c were obtained using the logarithmic form corresponding to the 21 sets of data selected in Table 4.

The mathematical models of the three milling components are:(4)$$F_{x}=10^{-0.1984}\cdot {v_{c}}^{0.2407}\cdot f^{0.7102}\cdot {a_{p}}^{0.6083}\;F_{y}=10^{-3.9666}\cdot {v_{c}}^{0.9488}\cdot f^{1.7896}\cdot {a_{p}}^{0.7658}\;F_{z}=10^{-3.8375}\cdot {v_{c}}^{1.8830}\cdot f^{0.8885}\cdot {a_{p}}^{-0.0869}$$

Moreover, the model was processed by variance analysis, and the $R^{2}$ of the three-direction milling forces were 0.92, 0.91, and 0.98, respectively. It suggests that the predicted value of the obtained milling force prediction model is consistent with the measured data, which indicates that the empirical force model is suitable for predicting the milling force of SiCp/2009Al. The *F* values were 80.26, 98.99, and 304.11, respectively. From the parameter table, it is found that $F_{0.01}\left(3,28\right)=4.68$ and the *F* values of the three-direction milling force are all greater than 4.68, which indicates that the regression model is accurate and has a certain significance.

To further verify the applicability of the developed milling force model in the actual cutting process, the regression model predicted milling forces using the nine test results in Table 4 described above. The measured milling forces for these nine data sets were compared with the theoretical values predicted by the milling force model, as shown in Figure 3.

According to the calculations, the average error of predated milling force is 13.47% in the x-direction, 6.58% in the y-direction, and 7.81% in the z-direction for the nine sets. The results demonstrate that the predicted error is in the reasonable range, and the model is verified. It will offer guidance for obtaining the best parameter combination for the optimization of milling integrity.

### 3.3. Sensitivity Analysis

To analyze the sensitivity of milling force $F_{total}$ to the parameters, range analysis was utilized. The analysis results are shown in Table 5. The results indicate that the influence of the three factors on the milling force is: milling speed > feed rate > milling depth.

The graph of milling speed and milling force is shown in Figure 4a. It can be seen from the figure that the milling force increases with the increase in milling speed, which is different from the traditional cutting theory. In the traditional cutting theory, the cutting force generally decreases with the increase in the cutting speed [20]. Zhou’s [21] milling tests of SiCp/Al composites showed that with increasing cutting speed v= 5–250 m/min, the three-way cutting force increases gradually. When the cutting speed exceeds v = 250 m/min, the cutting force tends to decrease. This change is related to the physical properties of the SiCp/Al composites themselves. This experiment was conducted under the conditions of low milling speed ($v_{c}\;<\;130\;m/\min$). On the one hand, as the milling speed increases, the friction angle changes insignificantly, and the chip inertia force increases; on the other hand, the increase in milling speed makes the cutting deformation decrease, the shear angle increases, and the shear force decrease to some extent. However, the degree of increase in chip inertia force is more obvious than the degree of decrease in shear force, and as a result, the milling force is increased. In addition, at low milling speeds, the cutting temperature is not sufficient to cause a softening effect on the aluminum substrate, which hardly contributes to the reduction in the milling force. Moreover, due to the specificity of the material, as the milling speed increases, the chance of collision between the tool and the particles increases, and the tool wears severely, causing the milling force increase, so in general, the milling force of SiCp/Al composites increases as the milling speed increases at lower milling speeds.

The graph of feed rate and milling force is shown in Figure 4b, and the milling force increases with the increase in feed rate. This is due to the increase in milling thickness caused by the increase in feed per tooth, which causes the normal force on the front tool face to increase, causing the force to rise rapidly. At the same time, the increase in feed per tooth increases the volume of material removed per unit time, which leads to an increase in the power required for cutting. This leads to an increase in the power required for cutting so that the cutting force also increases [22].

It is shown from Figure 4c that the milling force increases with milling depth, which is the relative position between the tool and the SiC particles. Milling depth is positively associated with SiCp/Al removal rate by affecting chip volume. Following that, the plastic deformation and cutting forces increase with increasing cutting edge-SiC particle interaction and chip-tool friction.

## 4. Removal Mechanism of SiCp/Al Composites

Since the cutting force is closely related to the actual state of material removal, the study of cutting force can help to reveal the removal mechanism of SiCp/Al composites. Under the milling parameters of a milling speed of 125.66 m/min (rotational speed of 4000 r/min), a feed rate of 320 mm/min, and a milling depth of 0.4 mm, the variation of three-way milling force with time was measured by a force measuring instrument, as shown in Figure 5. In the milling process, the miller passes over a large number of randomly distributed silicon carbide particles, causing them to collapse and fracture. The milling cutters move from softer aluminum substrates to harder silicon carbide particles, causing a sudden change in cutting force.

As shown in Figure 5, the force signal cutting force is choppy as milling progresses, which indicates that the material removal form is complicated. This suggests that during the cutting operation, the method of material removal is not constant. Since there are two phases in the SiCp/Al composite and the aluminum matrix is pliable while the SiC particles are brittle. As a result, the SiC particles and aluminum matrix should be studied separately from the removal mechanism of the SiCp/Al composites.

### 4.1. Removal Form of Aluminum Matrix

In the milling process of SiCp/Al composites, the removal mechanism of the aluminum matrix is very different from that of ordinary aluminum alloys because of the addition of SiC particles to the matrix.

Since the reinforced aluminum alloy is a typical plastic material, dislocation slip will occur during cutting due to shear stress. According to the Orowan dislocation bypass mechanism in classical fracture mechanics, if the SiC particle is small, the dislocation circumvents the SiC particle the slip line meets the SiC particle (the dislocation curvature radius R is half the distance between particles), as shown in Figure 6. In this case, the aluminum matrix will elastoplastic deformation during milling, resulting in plastic debris.

If the SiC particles encountered during dislocation slip in the matrix are large, the dislocation motion cannot bypass the SiC particles. Instead, it accumulates near the SiC particles. This leads to stress concentration.

Under the action of shear stress *τ*, the dislocation radius of curvature of the aluminum matrix is *R*:(5)$$R=\frac{G\;B}{2\tau }\;$$
where, G is the elastic modulus of the aluminum matrix, and B is the Burg vector.

The average spacing between particles is assumed to be $D_{p}$. When $R=D_{p}/2$, the dislocation bypass mechanism starts, and the aluminum matrix undergoes plastic deformation. At this time, the shear stress is the yield strength of the aluminum matrix of the SiCp/Al composite $\tau_{y}$:(6)$$\tau_{y}=\frac{G\;B}{D_{p}}\;$$

Therefore, during the milling process, when the shear stress $\tau >\tau_{y}$, applied to the SiCp/Al composite, the aluminum matrix undergoes plastic deformation

### 4.2. Removal Form of SiC Particles

According to the above analysis, elastoplastic deformation will occur in the process of the material milling removal process. However, the removal of SiC particles is complicated. The fracture of SiC particles is mainly found in three cases: (1), Interfacial debonding, pulling or pressing particles out of the matrix. (2), Particles are directly sheared off. (3), Particle crushing.

During milling, the milling force of the tool loaded onto the SiCp/Al composite material does not directly affect the SiC particles, but the force on the substrate is transferred to the SiC particles through the interface, reinforcing the force on the particles as shown in Figure 7a. Therefore, the form of particle removal during milling is also related to the strength of the interface.

In SiCp/Al composites, the interface between the matrix and particle is both weak (weak interface) and strong (strong interface). The weak interface is shown in Figure 7b. During the milling process, the weak interface between the particle and matrix is prone to interface crack, which causes the whole particle to overturn, flow out, or press in, forming a cavity and crack. The strong interface is shown in Figure 7c. When the interfacial strength is greater than the fracture strength of the silicon carbide particles, the matrix effectively transfers the load to the SiC particles through interfacial shear and can shear the SiC particles directly.

The surface morphology of the different removal forms of SiC particles is shown in Figure 8. Figure 8 shows the surface morphology of silicon carbide under different removal conditions. Figure 8a is a case of weak interfaces where interface disruption leads to the SiC being pulled out of holes, resulting in poor processing surface quality. Figure 8b shows the strong interface at which particles are sheared directly before the interface breaks. In this case, the processing surface quality is better. From Figure 8c, the visible fragmentation of the SiC particles results in the uneven surface quality of processing.

### 4.3. Influence of Milling Parameters on the Form of Material Removal

According to the foregoing, when the cutting speed is low, the milling force increases with the cutting speed. However, the theory of high-speed cutting shows that when the cutting speed increases to a certain value, the cutting force decreases with the increase in cutting speed. The increase in speed increases the temperature in the cutting process and the grain size in the aluminum alloy matrix. According to the dislocation theory, stress concentration expression manifests itself:(7)$$\tau =n\cdot \tau_{0}\;$$
where n is the number of dislocation plugs. The larger the n, the larger the stress concentration. According to the dislocation theory, the number of dislocation packing products is expressed as:(8)$$n=\frac{\tau_{0}\;D}{G\;B}\;$$

On the one hand, the greater the grain size, the greater the stress concentration. On the other hand, due to the different thermal expansion coefficients of the SiC particles and matrix, the stress concentration around the silicon carbide particles is more pronounced with the increase in velocity and temperature. Profits are more likely to reach the fracture strength of the SiC particles. SiC particles can be cut directly before matrix deformation to ensure better surface quality. Therefore, choosing the milling speed is beneficial to improve the machining quality.

Considering the material removal form in terms of milling depth, according to the Bifano ductility domain cutting critical depth model [23]:(9)$$d_{c}=\beta \frac{E}{H}{\left(\frac{K_{C}}{H}\right)}^{2}\;$$
where E represents the elastic modulus of the material, $K_{C}$ represents the fracture strength of the material, and H represents the hardness of the material. The research shows that the ductile removal condition of hard and brittle materials is when the surface cracks of milling account for 10% and β = 0.15 at this time. Since the critical cutting depth is also related to the parameters of the tool, it is necessary to add the tool influence coefficient $K_{0}$ based on the model. The critical depth of the cut model is obtained as:(10)$$h_{cu}=0.15\;K_{0\;}\frac{E}{H}{\left(\frac{K_{C}}{H}\right)}^{2}\;$$

According to the grinding theory of hard and brittle materials, the maximum undeformed thickness:(11)$$a_{gmax}={\left(\frac{4\;v_{w}}{v_{s}\cdot N_{d}\cdot C}\sqrt{a_{p}/d_{e}}\right)}^{\frac{1}{2}}\;$$
where $v_{w}$ is the workpiece speed, $v_{s}$ is the cutting speed, $a_{p}$ is the cutting depth, and $d_{e}$ is the equivalent diameter of the grinding wheel, $N_{d}$ and C are cutting constants.

It can be seen from the above Equations (10) and (11) that for a particular material and tool, its critical depth of cut is determined, while the maximum undeformed thickness is related to the machining parameters. Properly increasing the milling speed and reducing the milling depth can reduce the maximum undeformed chip thickness so that the maximum undeformed thickness of the tool cutting is less than the critical depth of the cut ($a_{gmax}<h_{cu}$), and the hard and brittle materials are removed in the ductile domain. Which, after removal, results in better surface quality after processing.

The above theoretical analysis is verified by the literature [24]. The parameters of the milling experiments are $v_{s}\;$= 20.4–222.3 m/s, $a_{p}\;$= 2.5 μm, $d_{e}\;$= 118 mm. The critical cutting thickness of the material is 1.28 μm, as calculated by Equation (10). According to the calculation of Equation (11), the cutting speed changes from 20.4 to 222.3 m/s, and the maximum undeformed cutting thickness decreases from 9 to 0.9 μm.

According to Figure 9a, a large number of pits and cracks can be seen on the machined surfaces, with high surface roughness. At this time, the maximum undeformed thickness of the tool cutting is greater than the critical depth of the cut ($a_{gmax}>h_{cu}$)), and the material may be assumed to undergo brittle removal. As can be seen from Figure 9b, there are fewer surface defects and relatively good surface quality after processing. At this time, the maximum undeformed thickness of the tool cutting is less than the critical depth of cut ($a_{gmax}<h_{cu}$)), and plastic removal can be considered to have occurred.

## 5. Conclusions

(1).From the milling experiments results, it was found that milling speed, feed rate, and milling depth had a significant impact on milling forces, respectively. Due to the particularity of SiCp/Al composites, the milling force increases with increasing speed in the milling process. The milling force increases with increasing feed rate and milling depth. From the above discussion, it was found that the milling force is affected by milling speed > feed rate > milling depth.(2).Through orthogonal experiments, the relevant parameters of the milling force empirical formula are determined, and the SiCp/Al composite milling force empirical formula model is established. The average error of the *x*-axis milling force prediction model is 13.47%, the *y*-axis milling force prediction model is 6.58%, and the *z*-axis milling force prediction model is 7.81%. The prediction model error is small, so the empirical formula can predict the milling force well.(3).The removal mechanism of reinforced phase and aluminum matrix in SiCp/Al composites is discussed in this paper. Among them, the aluminum matrix undergoes elasto-plastic removal, while the removal of SiC particles takes the form of debonding, cutting, and crushing.(4).The effect of milling speed on the material removal form of SiCp/Al composites during high-speed milling was analyzed theoretically. Based on the dislocation theory, it was found that the higher the milling speed, the better the milling force machining quality. Furthermore, the analysis revealed that decreasing the milling depth is effective for obtaining a better surface quality though affecting the undeformed chip thickness.

## Figures and Tables

**Figure 1 micromachines-13-01687-f001:**
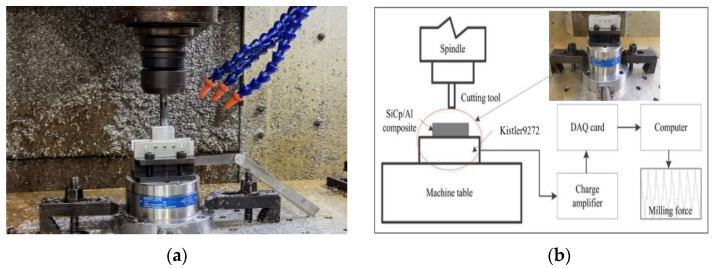
Milling site map and force measuring system (**a**) Milling experimental processing site map (**b**) Milling force measurement system diagram.

**Figure 2 micromachines-13-01687-f002:**
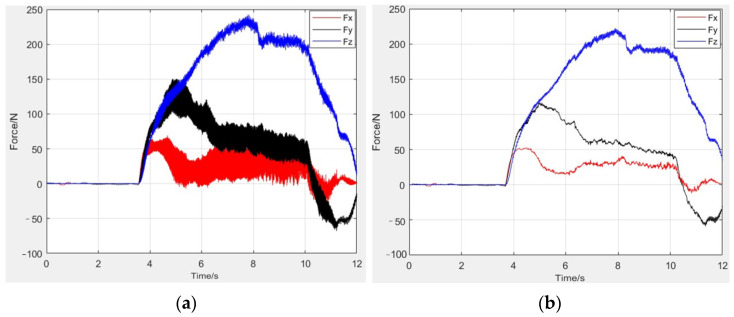
Milling force filtering, (**a**) Before filtering, (**b**) After filtering.

**Figure 3 micromachines-13-01687-f003:**
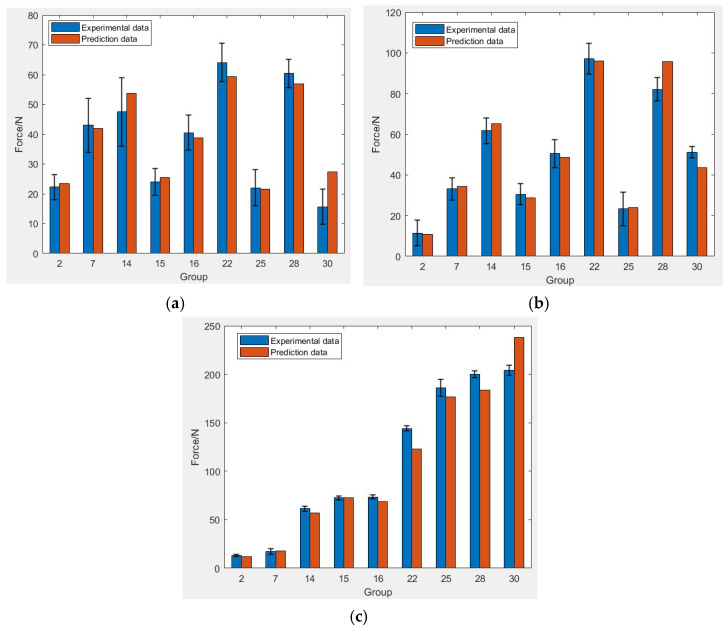
Comparison of experimental and predicted values of three-way milling force. (**a**) $F_{x}$; (**b**) $F_{y}$; (**c**) $F_{z}$.

**Figure 4 micromachines-13-01687-f004:**
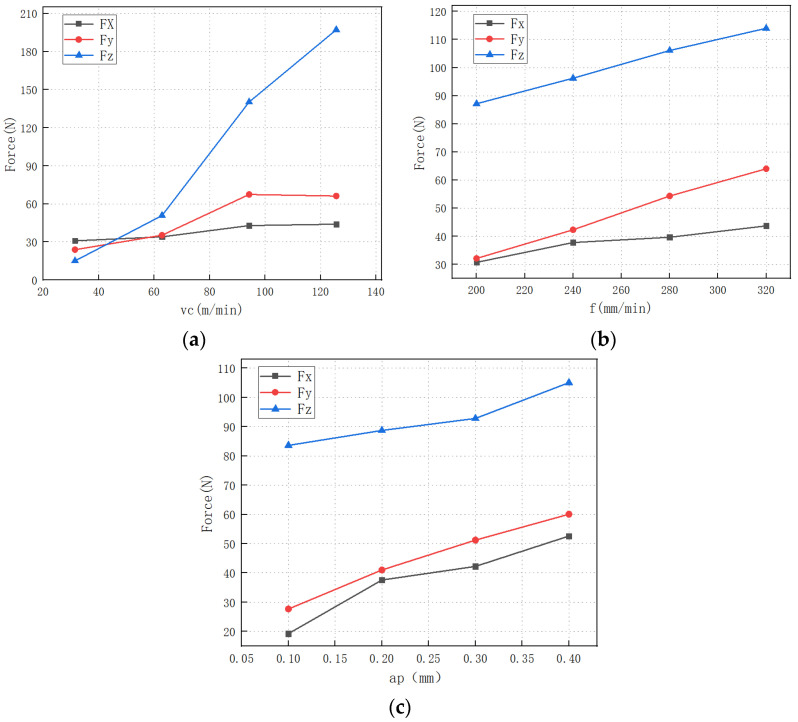
Variation of milling force with milling parameters. (**a**) milling speed; (**b**) feed rate; (**c**) milling depth.

**Figure 5 micromachines-13-01687-f005:**
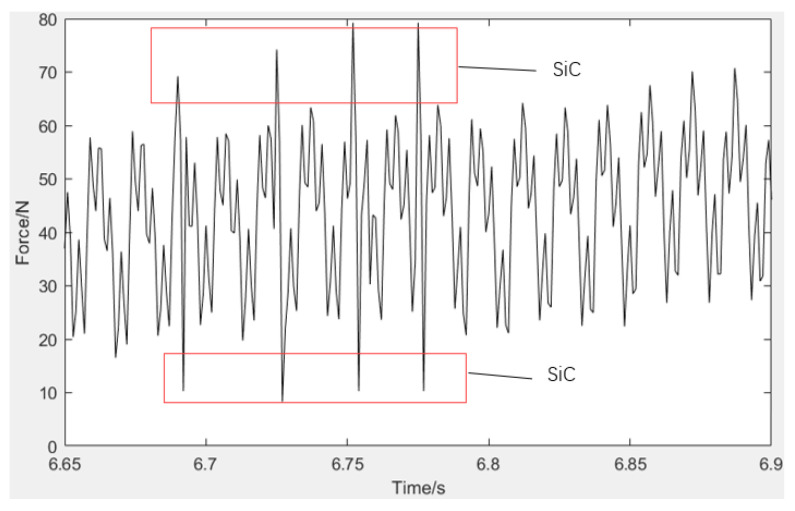
Variation curve of milling force under a single milling path.

**Figure 6 micromachines-13-01687-f006:**
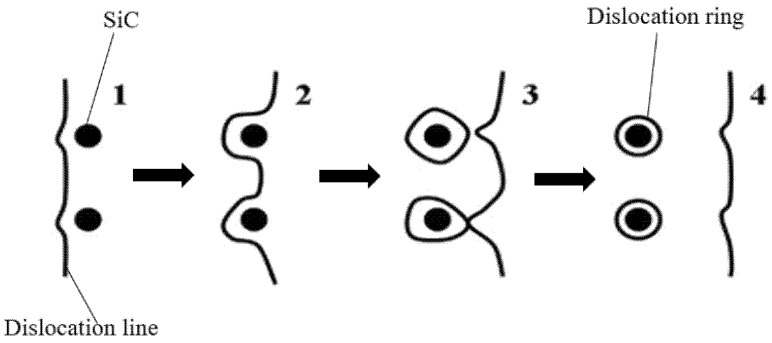
Dislocation bypass mechanism. (Dislocation shift process from 1 to 4).

**Figure 7 micromachines-13-01687-f007:**
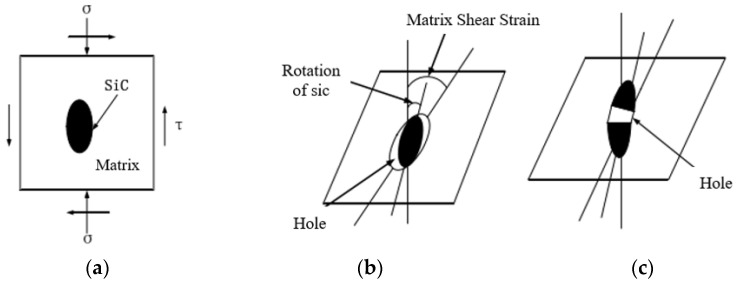
Schematic diagram of the removal method of SiC particles. (**a**) Force diagram of SiC; (**b**) low interface strength; (**c**) high interface strength.

**Figure 8 micromachines-13-01687-f008:**
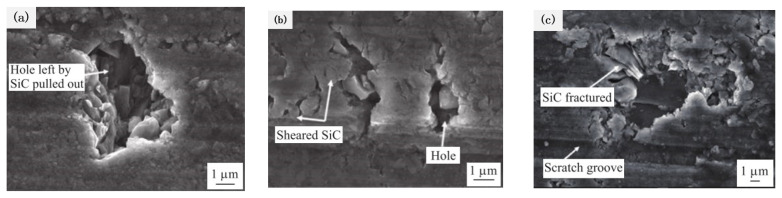
Surface morphology of milling SiCp/Al composites. (**a**). Hole left by SiC pulled out (**b**). Sheared SiC and Hole (**c**). SiC fractured and Scratch groove.

**Figure 9 micromachines-13-01687-f009:**
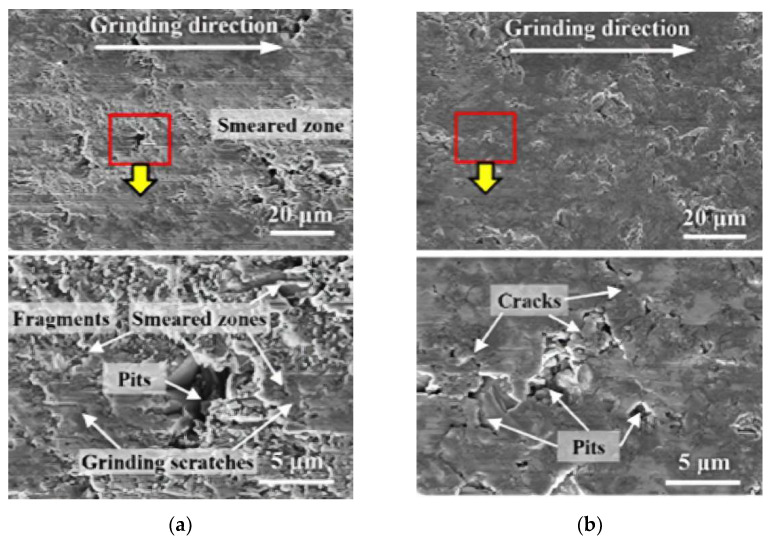
Material surface after machining at different cutting speeds. (**a**). $v_{s}\;$= 20.4 m/s, (**b**). $v_{s}\;$ = 222.3 m/s. Reprinted/adapted with permission from Ref. [24]. 2022, Elsevier Ltd.

**Table 1 micromachines-13-01687-t001:** Table of physical property parameters of SiCp/Al.

Symbol	Physical Quantity	Numerical Value
$\sigma_{b}$	Tensile strength	560 MPa
$\sigma_{s}$	Yield strength	380 MPa
Δ	Elongation	4–5%
HV	Hardness	200 HBS
E	Modulus of elasticity	111 GPa
α	Coefficient of thermal expansion	$16\times 10^{-6}$
ρ	Density	$2.85\;g/{\mathrm{cm}}^{3}$

**Table 2 micromachines-13-01687-t002:** Geometric parameters and mechanical properties of cutting tools.

Material	Modulus of Elasticity *E*/[MPa]	Density ρ$/[kg/m^{3}]$	Poisson’s Ratio v	Diameter *D*/[mm]	Cutter Tooth Number *Z*
YG6X	640,000	14,600	0.22	10	4

**Table 3 micromachines-13-01687-t003:** The orthogonal program of milling parameters.

Factors	$A\;\mathrm{Milling}\;\mathrm{Speed}\;v_{c}$ [m/min]	B Feed Rate f [mm/min]	$C\;\mathrm{Axial}\;\mathrm{Depth}\;\mathrm{of}\;\mathrm{Cutting}\;a_{p}\;[\mathrm{mm}]$
Level 1	31.42	200	0.1
Level 2	62.83	240	0.2
Level 3	94.25	280	0.3
Level 4	125.66	320	0.4

**Table 4 micromachines-13-01687-t004:** Orthogonal experimental scheme.

Number	A	B	C	$F_{x}\;$	$F_{y}\;$	$F_{z}\;$	$F_{total}\;$
1	A1	B1	C1	3.54	3.74	6.15	8.02
2	A1	B1	C2	22.28	11.45	13.22	28.32
3	A1	B2	C3	32.67	24.91	14.22	43.47
4	A1	B2	C4	42.86	33.69	12.72	55.98
5	A1	B3	C1	23.01	9.29	19.46	31.53
6	A1	B3	C2	32.09	21.01	19.83	43.18
7	A1	B4	C3	43.02	33.09	17.29	56.96
8	A1	B4	C4	47.87	53.89	18.01	74.30
9	A2	B1	C3	30.37	26.14	28.93	49.42
10	A2	B1	C4	39.69	35.25	34.21	63.15
11	A2	B2	C1	19.55	14.15	42.42	48.80
12	A2	B2	C2	31.35	24.53	42.51	58.24
13	A2	B3	C3	39.11	39.23	50.51	74.97
14	A2	B3	C4	47.53	61.71	61.52	99.26
15	A2	B4	C1	24.03	30.49	72.51	82.25
16	A2	B4	C2	40.56	50.55	73.54	98.02
17	A3	B1	C3	29.38	48.82	102.86	117.59
18	A3	B1	C4	54.25	67.39	133.76	159.30
19	A3	B2	C1	21.55	37.31	128.35	135.39
20	A3	B2	C2	41.97	60.74	133.54	152.59
21	A3	B3	C3	56.95	85.17	142.96	175.88
22	A3	B3	C4	64.09	97.13	144.25	185.34
23	A3	B4	C1	24.87	52.08	176.71	185.90
24	A3	B4	C2	49.98	90.39	159.95	190.40
25	A4	B1	C1	22.04	23.32	186.01	188.76
26	A4	B1	C2	44.03	40.78	191.97	201.13
27	A4	B2	C3	51.69	60.75	195.93	211.54
28	A4	B2	C4	60.39	82.14	200.09	224.57
29	A4	B3	C1	15.67	51.21	203.94	210.85
30	A4	B3	C2	38.59	69.83	206.07	220.98
31	A4	B4	C3	54.95	91.78	199.89	226.71
32	A4	B4	C4	64.31	109.56	194.07	231.95

**Table 5 micromachines-13-01687-t005:** $F_{total}$ range analysis results.

$F_{total}$	K1	K2	K3	K4
$v_{c}$	341.76	574.11	1302.39	1716.49
*f*	815.69	930.58	1041.99	1146.49
$a_{p}$	891.50	992.86	956.54	1093.85
Degree of influence	$v_{c}$ > *f* > $a_{p}$			
Optimal combination	$v_{c}$ = 125.66, *f* = 320, $a_{p}$ = 0.4			

## Data Availability

The datasets generated and analyzed during the current study are available from the corresponding author on reasonable request.

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
