# Peer review of "Force Prediction and Material Removal Mechanism Analysis of Milling SiCp/2009Al"

_micromachines, 2022, doi:10.3390/mi13101687_

Round 1

Reviewer 1 Report

In the paper entitled "Force prediction and material removal mechanism analysis of milling SiCp/2009Al", the authors established the empirical force model of milling SiCp/2009Al based on the multiple linear regression analysis. Furthermore, the material removal mechanism of SiCp/Al milling was also analyzed theoretically. Before acceptance, please consider the following comments.

1. So far, many prediction models for SiCp/Al composites cutting force have been established based on various algorithms. Therefore, what is the scientific value, research gap, novelty, originality and contribution of this paper. These need to be clarified carefully in the ‘Introduction’, and be logical.

2. Written English needs thorough checking. There are several excessively long sentences in the text and they should be broken up. For example, the sentence on page 7, lines 211-215. Furthermore, on page 3-line 100 and page 7-line 214, there's an extra full stop. On page 9-line 250, the last sentence uses the Chinese full stop. Please pay attention to these writing details!

3. On page 7-Figure 4, the pictures could be more prettier.

4. Figure 4(a) is very clear but the associated explanation is too superficial for such complex phenomena. In the article, the milling force increases with the increase of milling speed during SiCp/Al machining. This phenomenon contradicts some studies. Therefore, it must be proved by more detailed observations.

5. Why is the value below 130 m/min considered as low speed cutting? How is this critical value specified?

6. It was expressed in the text: the cutting force can change due to the presence of hard SiC particles. Where? Please mark it in Figure 5 if possible.

7. On page 8-line 230, is it wrong to write Figure 5 instead of Figure 6?

8. On page 9-line 242, what is ‘the shear stress curvature radius R’? Is that a proper noun? Please list specific references if available.

9. In the part of ‘Removal form of aluminum matrix of SiCp/Al composites’, does not the aluminum matrix of SiCp/Al composites undergo elastic-plastic deformation all the time during processing? So why should we analyze it in two cases? And the meaning of τy symbol was not explained in the text.

10. In the theoretical analysis, the authors have done the theoretical analysis of milling speed and milling depth. The conclusion also includes the milling feed, so where is the theoretical analysis of milling feed?

11. How does the authors verify the theoretical analysis? Please verify. The surface morphology or particle removal behavior of SiCp/Al composites under different conditions (For example, under the conditions of different cutting speed and different SiC-Al interface strength mentioned in the article) have not shown in this paper.

12. The theoretical analysis of critical cutting depth (maximum undeformed thickness) was presented sufficiently. However, the highest critical cutting depth is required to be at the micron level, but the authors' test parameters can completely ignore the critical depth of cut. So, what is the significance of this part of work for the thesis?

13. The conclusion is too long, too complicated, and the problems being addressed, need for the work or the outcomes achieved are not properly defined.

Author Response

请参阅附件。

Reviewer 2 Report

1. Repetition of punctuation marks. For example,

Although SiCp/Al composites are becoming more ... while enhancing its properties..

The influence of milling parameters ... of milling parameters on the milling force..

2. Please correct the punctuation in the quoted part of the original text.

Pramanik et al.[10],proposed a cutting force model ...should be modified to Pramanik et al. [10] proposed a cutting force model ...

3. In the introduction section, there is an ambiguity in Multiple regression analysis is used to establish the milling force. prediction model.. Please correct it.

4. The introduction section is too long, and references 6, 7, 8, and 9 are not very relevant. It is suggested to delete the introduction appropriately and keep the main literature.

5. As shown in Figure 1” should be added to the original text.

6. The selection of processing parameters are all relatively large, which does make the test phenomenon more pronounced. Please explain the basis for selecting the processing parameters in this way.

7. Are the horizontal coordinates of Figure 2 in seconds? Is Figure 2 a picture of the whole milling process? We generally take the smooth section of the curve during processing to extract the force. Because the machining parameters are large and the workpiece size is small, the horizontal coordinates scale has to be more detailed to make the curve trend more obvious.

8. The orthogonal test with three factors and four levels requires just 16 groups of tests, and more tests do fit the regression equation better. Why is it expanded according to SPSS software, and is there any basis here?

9. The equation for the milling force is more convincing when validated with data other than the fitted equation. R2 and F then indicate the rationality of the equation, so it would be repetitive to use the original data for validation. It is suggested to delete picture 3 and the exposition of picture 3. You can also keep this section if you think it is important.

10. Is there a mutation of force in Figure 5? Which part of the curve is the abrupt change in force when cutting SiC particles.

11. It can be seen from Figure 6 that the cutting force varies significantly as the cutting progresses, which indicates that the material removal form is not static during the cutting process.Is this seen in Figure 6? Please describe Figure 6 in detail.

12. Which part of Figure 6 is the SiC particle. Are the four cases in Figure 6 all dislocation mechanisms? If not, please explain each of the four scenarios.

13. Figure 7(a) was described, and what do (b) and (c) respectively represent, and also in words.

14. Figure 8(d) represents which case? Please describe Figure 8(d).

15. Please divide the fourth part into small sections. 4.1 4.2 ...

16. Conclusion (1) has a language repetition part.

17. In conclusion (3)

Based on the dislocation theory, it is found that the higher the milling speed, the smaller the milling force. This is in contradiction with our orthogonal test that the higher the milling speed the higher the milling force. “The graph of rotational speed and milling force is shown in Figure 4 (a). It can be seen from the figure that the milling force increases with the increase of rotational speed, which is different from the traditional cutting theory. ”

18. Does “Rotation speed” mean “milling speed”? I suggest using technical terms for processing parameters would be better.

19. What is the difference between high speed milling and low speed milling? With both low-speed and high-speed milling appearing in the conclusion, I think I have mixed up some of the conditions of the conclusion.

20. Line333milling speed > feed > milling depth. feed should be feed rate

21. How is the grinding mechanism verified?

22. The language of all the studies should be checked and polished.

Author Response

请看附件

Reviewer 3 Report

The authors presented a Force prediction and material removal mechanism analysis of milling SiCp/Al. This reviewer finds the manuscript is well-organized and technically sound. The reviewer recommends to accept the article in its current form.

Based on dislocation theory and max- 21 imum undeformed thickness theory, the effect of cutting parameters on the form of material re- 22 moval was analyzed, which serves a guide for selecting appropriate machining parameters to obtain 23 improved machining quality of SiCp/Al composites.

I think the authors could provide and few more references relevant to the work, specifically the work published in the last 5 years.

Round 2

Reviewer 1 Report

The author carefully revised and improved the manuscript according to the review comments, and agreed to accept the manuscript.

Reviewer 2 Report

no comment